# *HRAS* Mutations in Head and Neck Carcinomas in Japanese Patients: Clinical Significance, Prognosis, and Therapeutic Potential

**DOI:** 10.3390/ijms26073093

**Published:** 2025-03-27

**Authors:** Hidemi Ohshima, Eiji Kobayashi, Manabu Inaba, Ryotaro Nakazawa, Nobuyuki Hirai, Takayoshi Ueno, Yosuke Nakanishi, Kazuhira Endo, Satoru Kondo, Makiko Moriyama-Kita, Hisashi Sugimoto, Tomokazu Yoshizaki

**Affiliations:** Department of Otolaryngology-Head and Neck Surgery, Graduate School of Medical Science, Kanazawa University, Kanazawa 920-8640, Japan; hohshima@med.kanazawa-u.ac.jp (H.O.); inabamanabu@med.kanazawa-u.ac.jp (M.I.); ryo.yoroshiku@med.kanazawa-u.ac.jp (R.N.); nhhira@med.kanazawa-u.ac.jp (N.H.); uenotaka@med.kanazawa-u.ac.jp (T.U.); nakanish@med.kanazawa-u.ac.jp (Y.N.); endok@med.kanazawa-u.ac.jp (K.E.); ksatoru@med.kanazawa-u.ac.jp (S.K.); mkita@med.kanazawa-u.ac.jp (M.M.-K.); sugimohi@med.kanazawa-u.ac.jp (H.S.); tomoy@med.kanazawa-u.ac.jp (T.Y.)

**Keywords:** *HRAS* mutation, head and neck carcinoma, distant metastasis, cell migration, tipifarnib

## Abstract

It is well known that a number of head and neck carcinomas are associated with *HRAS* mutations, and that several cancers with *RAS* mutations, such as lung cancer, have a poor prognosis. In this study, we evaluated the frequency of *HRAS* mutations in head and neck carcinomas and characterized the clinical and cell biological features of carcinomas with *HRAS* mutations. *HRAS* mutations at codons 12, 13, and 61, mutational hot spots, were evaluated in tissue specimens obtained from 119 Japanese patients treated at our institution. DNA was successfully extracted from 100 specimens, and sequencing was completed. An *HRAS* mutation was found in 8 (8.0%) cases: 5 (6.1%) out of 82 HNSCCs and 3 (16.7%) out of 18 salivary gland carcinomas. Mutations were found at codons 12 and 61, while none were found at codon 13, which differs from previous reports. The mutation-positive cases had a relatively poor prognosis, consistent with previous reports, and were more frequently accompanied by distant metastasis. *HRAS* knockdown with siRNA suppressed the in vitro migration ability of *HRAS* mutation-positive cells but not that of *HRAS* mutation-negative cells. In conclusion, a positive *HRAS* mutation could be an indicator of distant metastasis and poor prognosis, as well as a potential therapeutic target.

## 1. Introduction

The prevalence of head and neck carcinoma (HNC) is increasing, making it the sixth most common cancer in the world (per the latest Global Cancer Observation, 2020). HNC includes malignancies arising from various anatomical sites, such as the oropharynx, nasopharynx, hypopharynx, larynx, oral cavity, and salivary glands. Despite remarkable advancement in cancer treatment, the prognosis for HNC remains poor [1,2]. One suspected reason why advancement in treatment of head and neck cancers is limited is that treatment modalities are not optimized for each specific cancer subset. There are several pivotal studies for HNC; however, these include patients with only a limited range of primary sites [3,4,5]. These head and neck squamous cell carcinoma (HNSCC) trials are based on anatomical site. The recent prevalence of categorizing cancers is based on genomic features, and the development of anticancer drugs has been shifting in this direction.

Recent advances in genetic analysis technology have led to the development of genome-precise medicine, and therapeutic agents targeting molecules related to driver genes have been put to practical use in various cancers, including lung cancer [6]. Genome-precise medicine has dramatically improved prognosis [7,8,9]. In fact, the NCCN guidelines on non-small-cell lung cancers recommend performing molecular testing to investigate druggable driver gene alterations before systemic drug therapy. However, the application of molecular-targeted drug therapy in HNCs is still in the exploratory stage.

The *RAS* family of small GTPases comprises three genes in humans, namely, *HRAS*, *NRAS*, and *KRAS*, which encode four proteins, HRAS, NRAS, KRAS4A, and KRAS4B, with the latter two KRAS isoforms arising from alternative splicing [10]. RAS proteins are involved in various biological responses, including cell proliferation, migration, growth arrest, senescence, differentiation, apoptosis, and survival. Patients with cancers with *RAS* mutations are likely to have a poorer prognosis than those with wild-type *RAS* [11]. In fact, as reported for lung cancer, colorectal cancer, and malignant melanoma, HNCs with positive *RAS* gene mutations have been reported to have a poor prognosis [11,12,13,14,15].

All RAS isoforms are substrates for farnesyltransferase, which catalyzes the binding of farnesyl groups to RAS proteins, enabling them to localize to the cell membrane and initiate oncogenic activity. Thus, farnesyltransferase inhibitors (FTIs) were considered potential therapeutic agents against *RAS*-mutated cancers, and several clinical trials were performed [16,17]. However, none proved to be effective. The reason FTIs do not have the anticipated effects is that KRAS and NRAS can be rescued from membrane displacement in the presence of FTIs by an alternative prenylation pathway, geranylgeranylation, catalyzed by geranylgeranyltransferase [18,19]. However, this is not the case with HRAS. HRAS is not a substrate for geranylgeranyltransferase; its membrane localization and cellular function are dependent on farnesylation and may be more suppressed by FTIs. *HRAS*-mutated cancers are still potential therapeutic targets of FTIs such as tipifarnib [15,20] despite the negative results of FTIs against *NRAS*- or *KRAS*-mutated cancers.

*HRAS* is the predominant mutated *RAS* isoform in several types of squamous cell carcinomas, including HNSCCs [10]. The Cancer Genome Atlas (TCGA) has reported that *HRAS* is mutated in 6% of HNSCCs at initial diagnosis [21]. It has been reported that patients with *HRAS*-mutant HNSCCs have poor clinical outcomes [15,22], and tipifarnib has demonstrated encouraging efficacy in patients with recurrent and/or metastatic HNSCCs with *HRAS* mutations, for whom therapeutic options are limited [20]. The situation in salivary gland carcinomas, which form a distinct subset apart from HNSCCs, is even more challenging. No standard treatment has been established in phase III clinical trials for salivary gland carcinomas, and molecular-targeted therapies, such as trastuzumab for HER2-positive tumors [23], have only been used in clinical practice. As the positivity rate of an *HRAS* mutation is relatively frequent in salivary gland carcinomas, ranging from 11.1 to 21% [24,25,26,27], FTIs can be considered one of the candidates for molecular-targeted drugs. In this study, we evaluated the frequency of *HRAS* mutations in HNCs, including salivary gland carcinomas, and we revealed the clinical characteristics of *HRAS* mutation-positive cancers. In addition, we analyzed the underlying biological mechanisms related to prognosis and distant metastasis, which were likely to be associated with *HRAS* mutations, in terms of migration ability and cytotoxicity using squamous cell carcinoma cell lines.

## 2. Results

### 2.1. HRAS Mutations Are More Frequently Detected in Salivary Gland Carcinomas and Are Possibly Associated with Distant Metastasis

Genomic DNA was extracted from specimens of 119 patients, and *HRAS* gene sequencing was available in 100 cases. *HRAS* gene mutations were detected in 8 out of 100 cases of HNCs (8%), with 4 cases at codon 12 and 4 cases at codon 61 (Table 1). All *HRAS* mutations at codon 12 were G12S. There were no significant differences in age, stage, lymph node metastasis, or treatment modalities between the cases with and without *HRAS* mutations. The proportion of females was higher among the cases with *HRAS* mutations than among those without *HRAS* mutations. The frequency of distant metastasis at diagnosis was significantly higher in the cases with mutations than in the cases without mutations (37.5% vs. 6.0%; *p* < 0.01).

*HRAS* mutations were found in 6.1% of HNSCCs (Table 2). As shown in Table 2, the cases with *HRAS* mutations showed a significantly higher frequency of distant metastasis than those without *HRAS* mutations (44.0% vs. 3.9%; *p* < 0.05). The incidence of *HRAS* mutations at our institute is consistent with previous reports. However, no mutations were found at codon 13 despite previous reports stating that a codon 13 mutation is relatively prevalent in HNSCCs [15]. There was no difference in the frequency of *HRAS* mutations between p16-positive oropharyngeal carcinoma and any other oropharyngeal carcinomas (5.9% vs. 10.5%; *p* = 0.453).

*HRAS* mutations were found in 14% of salivary gland carcinomas (Table 3). Distant metastasis at diagnosis was observed in only one case of salivary gland carcinoma with an *HRAS* mutation. Our data correlate with those in previous reports on salivary gland carcinomas [24,26,27].

### 2.2. HRAS Mutations Are Associated with Poor Prognosis in Several Subgroups of Head and Neck Carcinomas

There was no significant difference in OS or PFS between the cases with and without *HRAS* mutations (Figure 1A,B). For HNSCCs, there were no significant differences in OS or PFS between the cases with and without *HRAS* mutations (Figure 1E,F), whereas salivary gland carcinomas with *HRAS* mutations had a significantly shorter OS than those without *HRAS* mutations (Figure 1G).

As the prognosis of oropharyngeal carcinoma is strongly affected by HPV status, or p16 status [28], we analyzed the p16-positive oropharyngeal carcinoma cases separately. Although no significant differences in OS or PFS were observed between the p16-positive oropharyngeal carcinomas with and without *HRAS* mutations (Figure 2A,B), the p16-negative oropharyngeal carcinomas with *HRAS* mutations showed a significantly shorter PFS than those without *HRAS* mutations (Figure 2C,D).

### 2.3. Suppression of HRAS Expression Inhibits the Cell Migration of HRAS Mutation-Positive Cells

To elucidate the molecular effects by which *HRAS* mutations are involved in distant metastasis, the relationship between *HRAS* expression and cell migration was analyzed using a scratch assay. Treatment with siRNA against the *HRAS* gene markedly suppressed HRAS protein expression in KNS-62, HSC-2, and VU147T (Figure 3A). Cell migration was evaluated via changes in the area of cell defects after scratch treatment, as described previously [29] (Figure 3B–D). The suppression of *HRAS* expression significantly inhibited the migration of KNS-62 cells with *HRAS* mutations but not that of HSC-2 and VU147T cells without *HRAS* mutations. These data indicate that *HRAS* mutations play pivotal roles in the migration activity of carcinoma cells, which suggests that *HRAS* mutations also induce distant metastasis in actual patients.

### 2.4. Inhibition of HRAS Does Not Affect Susceptibility to Cetuximab and Cisplatin

The inhibitory effect of *HRAS* knockdown on the migration of KSN-62 cells might be explained by the inhibition of cell proliferation. Tipifarnib is a selective inhibitor of farnesyltransferase. The localization of HRAS to the cell membrane is exclusively dependent upon farnesylation, raising the possibility that tipifarnib functions as an HRAS inhibitor [19]. In this context, we examined the effects of *HRAS* knockdown and tipifarnib on cell proliferation or cytotoxicity. As shown in Figure 4, neither *HRAS* knockdown nor tipifarnib affected the viable cell numbers of KSN-62 cells or those of HSC-2 and VU147T cells.

Because *KRAS* mutations are involved in resistance to cetuximab in colorectal carcinoma [30,31], we next examined whether resistance to cetuximab and cisplatin, key anticancer agents for HNCs, is associated with *HRAS*. However, the dose-responsive cytotoxic effects of cetuximab and cisplatin were not affected by either *HRAS* knockdown or tipifarnib treatment.

## 3. Discussion

We demonstrated that patients with HNCs with *HRAS* mutations have a poorer prognosis, at least in some subgroups, and are more likely to have distant metastases at diagnosis than those without mutations. The migration of squamous cell carcinoma cells with an *HRAS* mutation was attenuated by the inhibition of HRAS protein expression, whereas those without an *HRAS* mutation were not affected. Although the precise function of HRAS in cell migration has not yet been clarified, the results suggest that HRAS is relevant to the complex metastatic system. However, irrespective of the *HRAS* mutation status, neither the suppression of HRAS protein expression nor tipifarnib treatment affected cell proliferation or sensitivity to cetuximab or cisplatin in vitro.

It has been reported that *HRAS* mutations are detected in 3–6% of HNSCCs [15,21]. Colemen et al. found that *HRAS* mutations occur more frequently in human papillomavirus-negative tumors, with *HRAS* G12C/D/F/N/S/V mutations being the most common (39.4%), followed by G13C/D/E/R/S/V/insG/insGGG (28%) and Q61 H/K/L/G (24%) [15]. In the present study, the frequencies of *HRAS* mutations in p16-positive and p16-negative oropharyngeal carcinomas were almost the same. *HRAS* mutations were distributed equally at the codon 12 and 61 sites, with no mutations at the codon 13 site. In our study, all patients were Japanese; thus, a difference in racial background is a possible reason for this discrepancy.

Salivary gland carcinomas form a subgroup which is quite different from HNSCCs. So far, there is no standard first-line chemotherapy for salivary gland carcinomas. They have a wide variety of histopathological types with rare incidences of squamous cell carcinomas and biological behaviors [32]. It is reported that epithelial myoepithelial carcinomas of salivary glands are characterized by a high incidence of *HRAS* mutations. Urano et al. reported that 81.7% of epithelial myoepithelial carcinomas harbor *HRAS* mutations [33]. Another report stated that detection of *HRAS* mutation was available for diagnosis of epithelial myoepithelial carcinomas [34]. However, *HRAS* mutations in salivary gland carcinomas are not limited to epithelial myoepithelial carcinomas. *HRAS* mutations are also detected in mucoepidermoid carcinomas, adenoid cystic carcinomas, acinic cell carcinomas, salivary duct carcinomas, and the incidence of *HRAS* mutations is reported to be 11.1–21%, which is relatively high compared to HNSCCs [24,25,26,27]. *HRAS* mutations of salivary gland carcinomas are detected in mucoepidermoid carcinomas and salivary duct carcinomas with an incidence of 14% of cases at our institute, which is consistent with previous reports. Saida et al. reported that salivary gland adenoid cystic carcinomas with *RAS* mutations have poorer prognosis compared to those with wild-type *RAS* [35]. We have also shown that salivary gland carcinomas with *HRAS* mutations have poorer OS. These findings suggest that salivary gland carcinomas with *HRAS* mutations form a subset of cases with poor prognosis. The fact that *HRAS* mutations are more prevalent in salivary gland carcinomas compared to HNSCCs, and are likely associated with poorer prognosis, suggests that *HRAS*-targeted therapy such as tipifarnib may hold significant promise not only for HNSCCs but also for salivary gland carcinomas. In fact, Hanna et al. have reported that tipifarnib showed modest clinical activity in a clinical trial [36].

A previous study reported that *HRAS* mutations were associated with poorer clinical outcomes in HNSCCs, and no difference in overall survival was observed between different *HRAS* codon mutations [22]. In our results, statistically significant differences in survival were observed in a limited number of stratified patients. However, both the OS and PFS of patients with *HRAS* mutations seemed to demonstrate a trend toward being shorter. This trend is consistent with previous reports [15,22]. One clear exception was observed in p16-positive cases. The OS of p16-positive cases was nearly the same regardless of *HRAS* status, and PFS seemed to be relatively longer in patients with *HRAS* mutations. Although factors affecting prognosis include the stage at initial diagnosis, the presence of distant metastases, and responsiveness to treatment, the relative importance of these factors remains unclear. Our data show that *HRAS* mutations are more likely to be associated with distant metastasis, and, among the HNC patients with distant metastases, those with *HRAS* mutations had a poorer prognosis than those without *HRAS* mutations. These results suggest the importance of distant metastasis for the prognosis of *HRAS* mutation-positive patients. Consistent with previous reports, cell migration involved in metastasis is associated with the presence of *HRAS* mutations [37]. Taken together, *HRAS* is possibly one of the driver genes in the carcinogenesis of HNCs. However, it may not function as a single definitive driver, as seen in lung carcinomas, nor as a single definitive prognostic factor, such as HPV status in oropharyngeal carcinomas. There could be co-factors other than *HRAS* mutations, such as other gene mutations.

PI3K and RAS are important regulators of cell motility and chemotaxis, and they influence each other’s activity through direct and indirect feedback processes [38]. *HRAS* knockdown in fibroblasts abrogates the local amplification of 3-PI signals upon synthetic PI3K activation and results in short-lived protrusion events that do not support cell migration in response to platelet-derived growth factor [26]. A significant inhibitory effect of *HRAS* knockdown on cell migration was observed in *HRAS* mutation-positive cells, but not in *HRAS* mutation-negative cells, suggesting that mutation-positive *HRAS* plays a pivotal role in the migration of squamous carcinoma cells, resulting in distant metastasis. *PI3K* mutations are also frequently found in salivary duct carcinomas [26,27], and the co-mutation of *HRAS*/*PI3K* has been reported [24,39]. The importance of *HRAS* mutations in cell migration may vary in each patient, depending on the presence of coexisting PI3 pathway abnormalities. Thus, combination therapy that inhibits both HRAS and PI3K seems to be a promising option. In fact, a phase I/II trial is ongoing in patients with recurrent or metastatic HNSCCs who have *PIK3CA* mutations in addition to *HRAS* mutations, combining tipifarnib with alperisib, a PI3K inhibitor (NCT04997902). The combination therapy of an FTI with a PI3K inhibitor has the potential to improve efficacy in patients who are refractory to FTI monotherapy.

In recent years, the therapeutic benefits of novel agents, such as immune checkpoint inhibitors, have been evaluated for the treatment of HNCs. However, cetuximab and cisplatin remain cornerstone drugs due to their potential for curative outcomes. In colorectal carcinoma, resistance to cetuximab has been observed in patients harboring *KRAS* mutations, a subset of *RAS* mutations [30,31]. Unfortunately, neither *HRAS* knockdown nor tipifarnib demonstrated significant effects on in vitro cytotoxicity or sensitivity to cetuximab and cisplatin. Further studies are needed to evaluate the effects of *HRAS* mutations on resistance to drugs such as cetuximab and cisplatin.

Our current study has several limitations. First, it was performed retrospectively with a relatively limited number of patients. The small sample size precludes definitive statistical conclusions. Second, sequencing data were not available for 19 out of 119 specimens, 16% of cases. This relatively high rate of unavailability may be due to the inadequate processing or preservation condition of the specimens. For example, the formalin fixation times may have been too long, or the specimens may have been too old. Additionally, specimen size was limited, especially the biopsied specimen size. Third, we examined the relationship between *HRAS* expression and cell migration or sensitivity to anticancer drugs in only three cell lines. In particular, the *HRAS*-mutated cell line was substituted by a lung cancer cell line with *HRAS* mutations, because the availability of the *HRAS* mutation-positive HNSCC cell line was limited. Further studies are needed to overcome these limitations in order to demonstrate, in greater detail, the mechanisms by which *HRAS* mutations play roles in the carcinogenesis of HNCs, such as the molecular pathway of HRAS signaling, and to discover subsets of patients who may benefit from treatment with FTIs, or adequate agents to administer along with FTIs, such as PI3K inhibitors. Despite these limitations, this study may have uncovered some interesting and clinically relevant results in terms of the characteristics of *HRAS* mutation-positive HNCs.

In conclusion, a positive *HRAS* mutation is possibly an indicator of distant metastasis and poor prognosis in HNCs. The inhibition of the *HRAS* mutation-positive protein is expected to be used in clinical settings, especially in HNSCC patients with distant metastases.

## 4. Materials and Methods

### 4.1. Patients and Specimens

Clinical specimens were obtained from 119 patients with HNCs (oropharyngeal, oral, and salivary gland carcinomas) who were initially treated at Kanazawa University Hospital between April 2018 and March 2023, and from whom sufficient tissue specimens were available. Genomic DNA was extracted from the specimens, and *HRAS* mutations were detected via Sanger sequencing (details are shown below). Of the 119 patients, 100 were included in the analysis, and 19 were excluded due to sequencing failure. The patients’ clinical data were obtained from medical records. This retrospective study was conducted with the approval of the Kanazawa University Ethics Committee (Approval No. 114666–1).

### 4.2. Cell Lines and Knockdown with siRNA

KNS-62, a lung squamous cell carcinoma cell line with a Q61L *HRAS* mutation, and HSC-2, an oral squamous cell carcinoma cell line, were obtained from the Japanese Collection of Research Bioresources (JCRB) cell bank (Osaka, Japan). VU147T, a p16-positive oropharyngeal squamous cell carcinoma cell line, was a gift from Dr. H. Joenje (VU Medical Center, Amsterdam, The Netherlands). KNS-62 was maintained in Minimum Essential Medium (MEM) (Thermo Fisher Co., Waltham, MA, USA), supplemented with 20% fetal bovine serum (FBS) (Thermo Fisher Co., Waltham, MA, USA) and HSC-2, and VU147T was cultured in Dulbecco’s modified Eagle medium (DMEM) (Thermo Fisher Co., Waltham, MA, USA), supplemented with 10% FBS at 37 °C in 5% CO_2_. The *HRAS* gene in the cell lines was knocked down using On-target plus Human *HRAS* siRNA or siGENOME Non-Targeting Control siRNAs (Horizon Discovery Co., Cambridge, UK) and Lipofectaimin RNAiMAX Transfectant Reagent (Thermo-Fisher Co., Waltham, MA, USA), according to the manufacturer’s instructions. *HRAS* knockdown was confirmed via Western blotting, as previously described [29,40]. In brief, transfected cells were lysed using RIPA buffer. The protein concentration of the cell lysate was determined using a Bio-Rad Protein Assay (Bio-Rad Laboratory Inc., Hercules, CA, USA). Next, 30 μg of protein was electrophoresed and transferred to nitrocellulose membranes. The membranes were then stained with an anti-b-actin antibody (Thermo-Fishier Co., Waltham, MA, USA) or an anti-HRAS antibody (Abcam Lim., Cambridge, UK) and incubated with the appropriate horseradish peroxidase-conjugated secondary antibodies (Bio-Rad Laboratory Inc., Hercules, CA, USA). Bands were visualized with an enhanced chemiluminescence reagent (Cytiva, Tokyo, Japan) and detected using ChemiDoc (Bio-Rad Laboratory Inc., Hercules, CA, USA).

### 4.3. DNA Extraction and Detection of HRAS Mutations

The paraffin-embedded specimens were sliced into two 10 μm thick, 1.5 × 1.5 cm^2^ slices by selecting as much tumor area as possible. After deparaffinization, genomic DNA was extracted using an AllPrep DNA/RNA FFPE Kit (Qiagen K.K. Venlo, The Netherlands). Fragments of the codon 12–13 and 61 regions, which are mutational hot spots of the *HRAS* gene, were amplified via PCR. All PCRs were performed with KOD One PCR Master mix (Toyobo Inc., Osaka, Japan). The primers used in the PCR for *HRAS* gene sequencing are as previously reported [40]: H-ras-12 H5′: 5′GAGACCCTGTAGGAGGACCC3′, H-ras-12 H3′: 5′GGGTGCTGAGACGAGGGACT3′, H-ras-61 H-6IS: 5′ATGAGAGGTACCAGGGAGAG3′, and H-ras-61 H-61A: 5′TCACGGGGTTCACCTGTACT3′. The Sanger sequencing reaction was performed using a BigDye Terminator Cycle Sequence Kit ver. 3 (Thermo-Fisher Co., Waltham, MA, USA). Examples of the detected *HRAS* mutations are shown in Figure 5.

### 4.4. Scratch Assay

The cell monolayer was scraped in a straight line to create a scratch wound with a p1000 pipet tip, as previously described [29]. Phase-contrast microscopic images were acquired periodically after cell culture. The area of cell defect in the wound was quantified by Photoshop (Adobe Inc., San Jose, CA, USA) and was expressed as a ratio to the area immediately after scratching. The percentage of the area was calculated using the following:wounded area (%) = (width after a certain time/width at beginning) × 100%.(1)

### 4.5. MTS Assay

A 3-(4,5-dimethylthiazol-2-yl)-5-(3-carboxymethoxyphenyl)-2-(4-sulfophenyl)-2H-tetrazolium salt (MTS) assay was performed to assess the effect of the HRAS inhibition on cell proliferation, as described previously [29]. In brief, KNS-62 and HSC-2 were grown at a concentration of 3 × 10^3^ cells/well and VU147T was grown at a concentration of 6 × 10^3^ cells/well, in 96-well plates. The cells were incubated for 24 h in the CO_2_ incubator. The cells were subsequently treated with serial dilutions of cetuximab (Merck KGaA, Darmstadt, Germany) or cisplatin (Nichi-iko Co., Toyama, Japan) in the presence or absence of 20 nM tipifarnib for 48 h. An MTS assay was performed, according to the manufacturer’s instructions. Briefly, 20 µL MTS (CellTiter 96 AQueous One Solution Cell Proliferation Assay, Thermo-Fisher Scientific, Waltham, MA, USA) was added to the cells and incubated for 2 h at 37 °C. Following MTS incubation, the spectrophotometric absorbance of the samples was estimated. Cells without the drug were used as controls. The percentage of viable cells was calculated using the following equation:(mean optical density (OD) of the experiment/mean OD of the control) × 100.(2)

### 4.6. Statistics

The results are expressed as means ± SD of the mean. Differences between two groups were analyzed using Welch’s non-paired *t*-test.

All statistical analyses were performed with SPSS ver. 29.0.2.0 (IBM, Armonk, NY, USA). The Mann–Whitney U test or Fisher’s exact probability test was used to compare the variables between the two groups. The data are presented as the median and range or the mean and standard deviation. Overall survival (OS) and progression-free survival (PFS) were analyzed using the Kaplan–Meier method, and the log-rank test was used to compare the patient groups. A *p*-value < 0.05 was considered statistically significant.

## Figures and Tables

**Figure 1 ijms-26-03093-f001:**
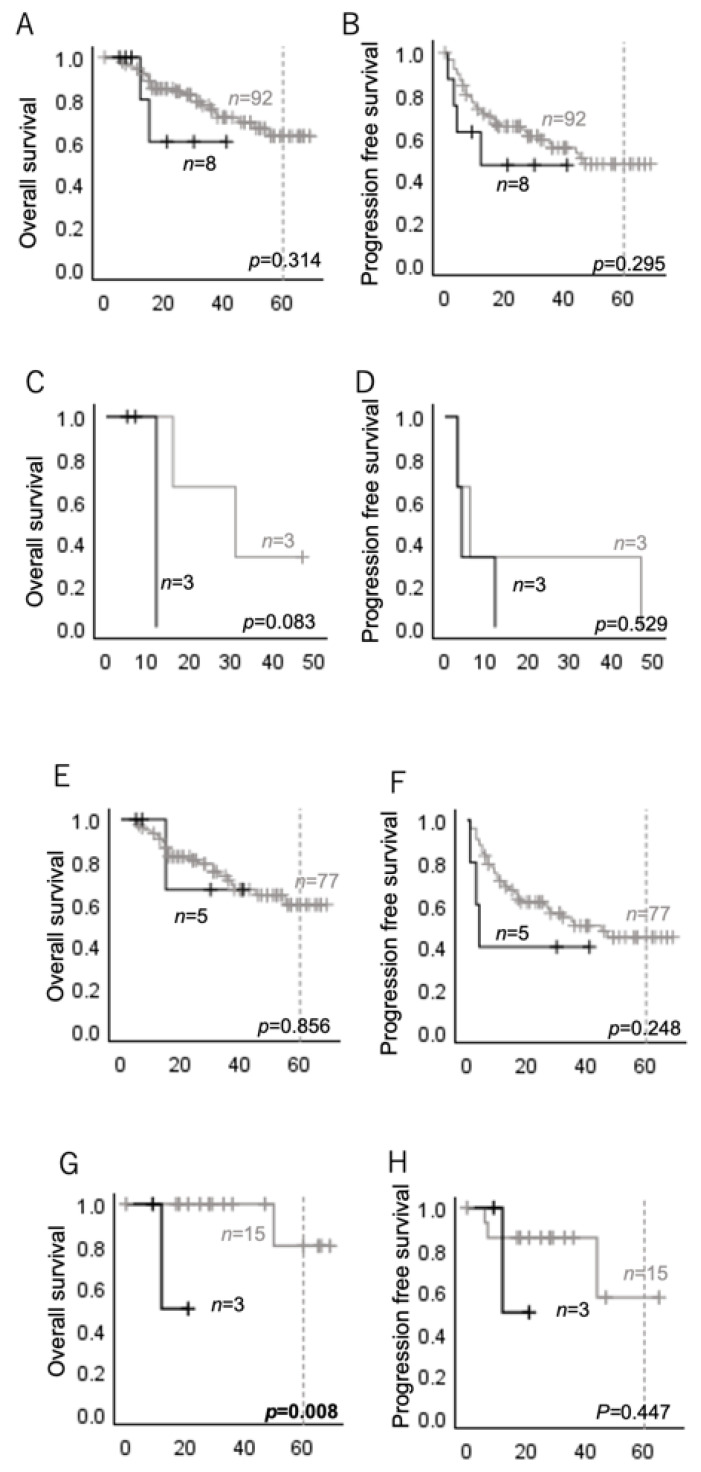
Overall survival (OS) and progression-free survival (PFS) according to the presence or absence of *HRAS* mutations were analyzed using a Kaplan–Meier analysis. Black lines indicate *HRAS*-mutated cases, and gray lines indicate *HRAS* wild-type cases. (**A**,**B**) OS and PFS of all head and neck cancers; (**C**,**D**) OS of HNCs with distant metastases; (**E**,**F**) OS and PFS of HNSCCs; (**G**,**H**) OS and PFS of salivary gland cancers.

**Figure 2 ijms-26-03093-f002:**
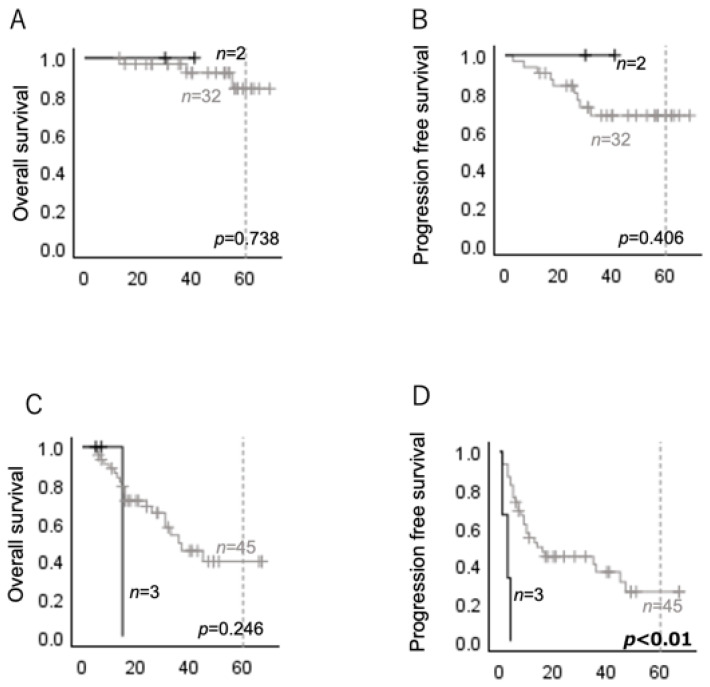
Overall survival (OS) and progression-free survival (PFS) according to the presence or absence of *HRAS* mutations were analyzed using a Kaplan–Meier analysis. Black lines indicate *HRAS*-mutated cases, and gray lines indicate *HRAS* wild-type cases. (**A**,**B**) OS and PFS of p16-positive oropharyngeal carcinomas; (**C**,**D**) OS and PFS of p16-negative oropharyngeal carcinomas.

**Figure 3 ijms-26-03093-f003:**
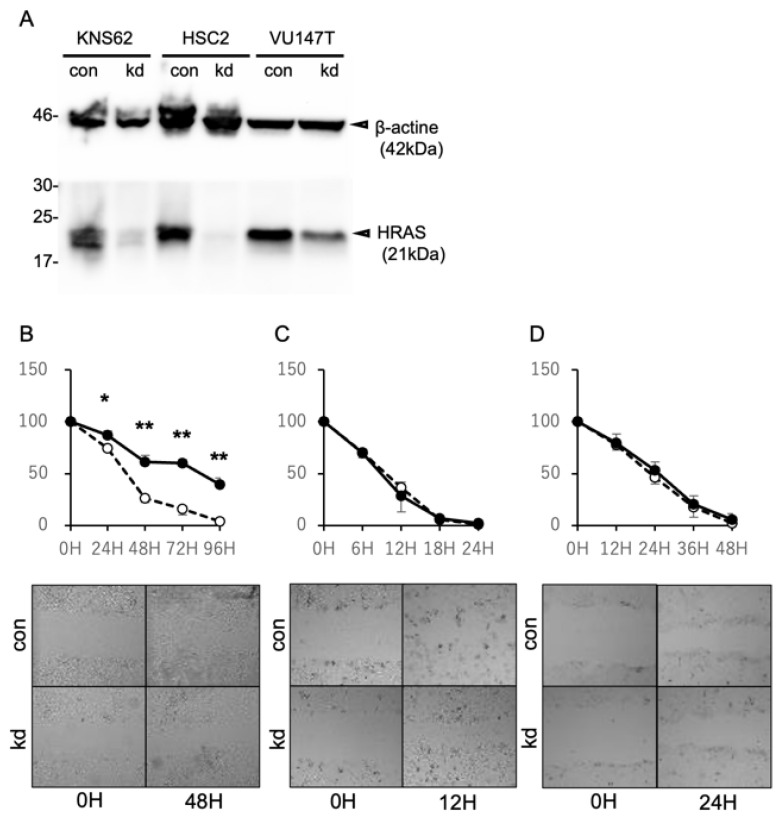
Effects of *HRAS* gene status on cell migration. (**A**) Expression levels of HARS protein on cell lines determined via Western blotting in siRNA knockdown (kd) and control cells; (**B**–**D**) cell migration evaluated using a scratch assay. The change in the width of defects of KNS-62 (**B**), HSC-2 (**C**), and VU147T (**D**) cells is expressed as a mean percentage. *HRAS* knockdown cells are indicated by closed circles, and control cells are indicated by open circles. Photos shown are representative results from each of the three independent experiments. * *p* < 0.05, ** *p* < 0.01. mean ± SD.

**Figure 4 ijms-26-03093-f004:**
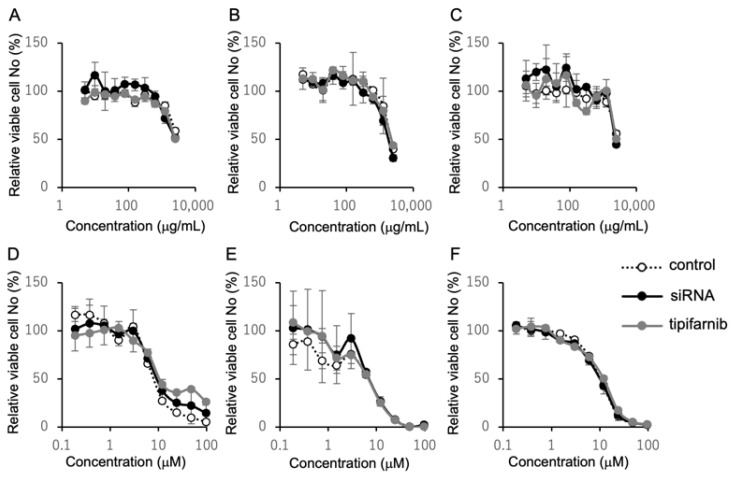
Effects of *HRAS* gene status on drug resistance against cetuximab or cisplatin. KNS-62 (**A**,**D**), HSC-2 (**B**,**E**), and VU147T (**C**,**F**) cells were treated with siRNA of *HRAS* (closed circles) or tipifarnib (gray circles) or not treated (open circles). The cells were cultured in the presence of the indicated concentrations of cetuximab (**A**–**C**) or cisplatin (**D**–**F**) for 48 h. Viable cell numbers were measured using an MTS assay and are indicated as a percentage of the numbers of viable cells in the absence of cetuximab or cisplatin. The results shown are the means of three independent experiments.

**Figure 5 ijms-26-03093-f005:**
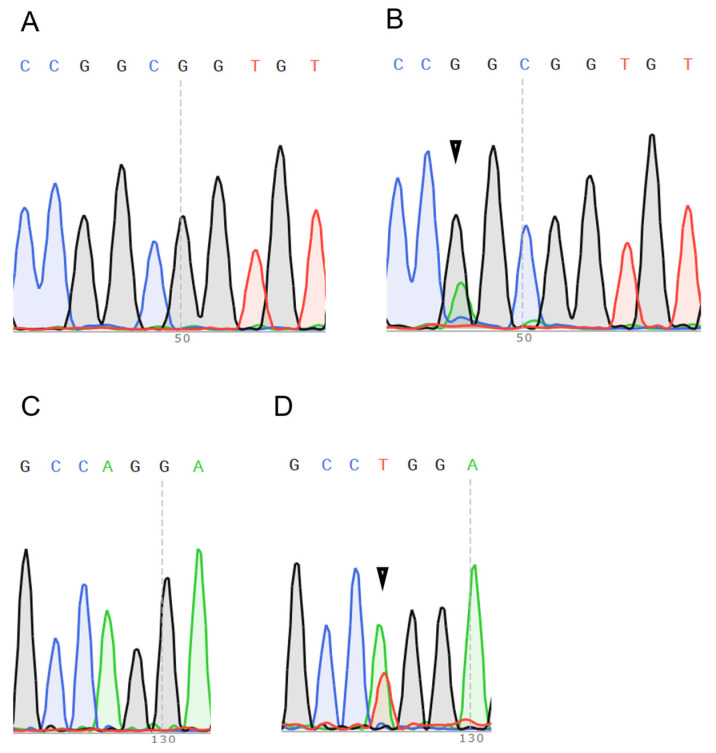
Wild type (**A**) and a point mutation at the first base (G to A) (**B**) of codons 12–13 (GGC to GGT). Wild type (**C**) and a point mutation at the second base (A to T) (**D**) of codon 61. Arrow indicates mutated base. Green: A; Black: G; Blue: C; Red: T.

**Table 1 ijms-26-03093-t001:** Characteristics of the study subjects. * *p* < 0.05.

	Total	Mutation		*p*-Value
		Positive	Negative	
**Numbers (%)**	**100**	8 (8.0)	92 (92.0)	
**Codon (%)**				
Mutation at codon12		4 (50.0)		
Mutation at codon13		0 (0.0)		
Mutation at codon61		4 (50.0)		
**Male:Female**	73:27	3:5	70:22	**0.032 ***
**Race** Asian	100	8 (100.0)	92 (100.0)	
**Age** median (range) years	70 (22–89)	70.5 (47–82)	70 (22–89)	0.746
**Primary site (%)**				0.286
Oral cavity	26 (26.0)	1 (12.5)	25 (27.2)	
Oropharynx	56 (56.0)	4 (50.0)	52 (56.5)	0.726
P16 positive	34	2	32	
P16 negative	19	2	17	
P16 unknown	3	0	3	
Salivary glands	18 (18.0)	3 (37.5)	15 (16.3)	
**Stage (%)** 1	29 (29.0)	3 (37.5)	26 (28.3)	0.431
2	24 (24.0)	0 (0.0)	24 (26.1)	
3	10 (10.0)	1 (12.5)	9 (9.8)	
4	37 (37.0)	4 (50.0)	33 (35.9)	
**Tumor stage (%)** 0	1 (1.0)	1 (12.5)	0 (0.0)	**0.008 ***
1	19 (19.0)	0 (0.0)	19 (20.7)	
2	33 (33.0)	2 (25.0)	31 (33.7)	
3	21 (21.0)	2 (25.0)	19 (25.0)	
4	26 (26.0)	3 (37.5)	23 (31.5)	
**Lymph node metastasis (%)**	60 (60.0)	7 (87.5)	53 (57.6)	0.14
**Distant metastasis (%)**	6 (6.0)	3 (37.5)	3 (3.3)	**0.006 ***
**Initial treatment (%)**	64 (64.0)	5 (62.5)	59 (64.1)	0.206
Operation	73 (73.0)	4 (50.0)	69 (75.0)	
Chemoradiation	27 (27.0)	4 (50.0)	23 (25.0)	
**Observation period median (range) months**	31 (0–69)	13.5 (5–41)	31.5 (0–69)	**0.011 ***

**Table 2 ijms-26-03093-t002:** Characteristics of the patients with squamous cell carcinoma. * *p* < 0.05.

	Total	Mutation		*p*-Value
		Positive	Negative	
**Numbers (%)**	**82**	5 (6.1)	77 (93.9)	
Codon (%)				
Mutation at codon12		3 (60.0)		
Mutation at codon13		0 (0.0)		
Mutation at codon61		2 (40.0)		
**Male:Female**	64:18	2:3	62:15	0.068
**Race** Asian	82	5 (100.0)	77 (100.0)	
**Age** median (range) years	69.5 (23–86)	71 (47–82)	69 (23–86)	0.572
**P16 positive**	34 (41.5)	2 (40.0)	32 (41.6)	1.0
**Stage (%)** 1	25 (30.5)	2 (40.0)	23 (29.9)	0.379
2	21 (25.6)	0 (0.0)	21 (27.3)	
3	8 (9.8)	0 (0.0)	8 (10.4)	
4	28 (34.1)	3 (60.0)	25 (32.5)	
**Tumor Stage (%)** 0	1 (1.2)	1 (20.0)	0 (0.0)	**0.002 ***
1	16 (19.5)	0 (0.0)	16 (20.8)	
2	29 (35.4)	1 (20.0)	28 (36.4)	
3	16 (19.5)	1 (20.0)	15 (19.5)	
4	20 (24.4)	2 (40.0)	18 (23.4)	
**Lymph node metastasis (%)**	53 (64.6)	5 (100)	48 (62.3)	0.156
**Distant metastasis (%)**	5 (6.1)	2 (40.0)	3 (3.9)	**0.028 ***
**Initial treatment (%)**				0.163
Operation	57 (69.5)	2 (40.0)	55 (71.4)	
Chemoradiation	25 (30.5)	3 (60.0)	22 (28.6)	
**Observation period median (range) months**	31 (5–69)	15 (5–41)	31 (5–69)	0.089

**Table 3 ijms-26-03093-t003:** Characteristics of the patients with salivary gland carcinoma. * *p* < 0.05.

	Total	Mutation		*p*-Value
		Positive	Negative	
**Numbers (%)**	**18**	3 (16.7)	15(83.3)	
**Codon (%)**				
Mutation at codon12		1 (33.3)		
Mutation at codon13		0 (0.0)		
Mutation at codon61		2 (66.7)		
**Male:Female**	9:9	1:2	8:7	1
**Race (%)** Asian	18	3 (100.0)	15 (100.0)	
**Age** median (range) years	71 (22–89)	70 (60–72)	72 (22–89)	0.654
**Primary site (%)**				1
Parotid gland	15 (83.3)	3 (20.0)	12 (80.0)	
Submandibular gland	3 (16.7)	0 (0.0)	3 (100.0)	
Sublingual gland	0 (0.0)	0 (0.0)	0 (0.0)	
Minor salivary gland	0 (0.0)	0 (0.0)	0 (0.0)	
**Histopathology (%)**				0.65
Squamous cell	2 (11.1)	0 (0.0)	2 (13.3)	
Myoepithelial	2 (11.1)	1 (33.3)	1 (6.7)	
Acinic cell	2 (11.1)	0 (0.0)	2 (13.3)	
Adenoid cystic	1 (5.6)	0 (0.0)	1 (6.7)	
Salivary duct	3 (16.7)	1 (33.3)	2 (13.3)	
Mucoepidermoid	4 (22.2)	1 (33.3)	3 (20.0)	
Secretory	4 (22.2)	0 (0.0)	4 (26.7)	
**Stage (%)** 1	4 (22.2)	1 (33.3)	3 (20.0)	0.457
2	3 (16.7)	0 (0.0)	3 (20.0)	
3	2 (11.1)	1 (33.3)	1 (6.7)	
4	9 (50.0)	1 (33.3)	8 (53.3)	
**Tumor stage (%)** 1	3 (16.7)	0 (0.0)	3 (20.0)	0.84
2	4 (22.2)	1 (33.3)	3 (20.0)	
3	5 (27.8)	1 (33.3)	4 (26.7)	
4	6 (33.3)	1 (33.3)	5 (33.3)	
**Lymph node metastasis (%)**	7 (38.9)	2 (66.7)	5 (33.3)	0.528
**Distant metastasis (%)**	1 (5.6)	1 (33.3)	0 (0.0)	0.167
**Initial treatment (%)**				0.314
Operation	16 (88.9)	2 (66.7)	14 (93.3)	
Chemoradiation	2 (11.1)	1 (33.3)	1 (6.7)	
**Observation period median (range) months**	28.5 (2–69)	12 (9–21)	33 (0–69)	**0.039 ***

## Data Availability

The data presented in this study are available on request from the corresponding author, as they contain personal patient information.

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
