# Peer review of "HRAS Mutations in Head and Neck Carcinomas in Japanese Patients: Clinical Significance, Prognosis, and Therapeutic Potential"

_ijms, 2025, doi:10.3390/ijms26073093_

Round 1
Reviewer 1 Report
Comments and Suggestions for Authors
Thank you for the chance to review the manuscript with the title “HRAS Mutations in Head and Neck Carcinomas in Japanese patients: Clinical Significance, Prognosis, and Therapeutic Potential” by Oshima et al..
The aurthors report the prevalence, prognostic significance, and therapeutic potential of HRAS mutations in head and neck carcinomas in a Japanese patient cohort. While the overall mutation rate was 8%, a higher rate of HRAS mutations among patients with salivary gland carcinomas (16.7%) compared to head and neck squamous cell carcinomas (HNSCCs, 6.1%) were reported by the authors.
The study found that HRAS mutations are associated with poorer prognosis, mainly associated with distant metastases. Although the difference of overall survival (OS) and progression-free survival (PFS) between HRAS mutant and wild type head and neck patients were not statistically significant, the stratified analysis provide the insights. Among the salivary gland carcinoma patients, those with HRAS mutations demonstrated significantly shorter OS compared to those who were HRAS wild-type, suggesting an aggressive disease course associated with HRAS mutation. Patients with p16-negative oropharyngeal carcinomas with HRAS mutations showed significantly shorter PFS, suggesting that HRAS mutations may associate with an accelerated disease progression in this subgroup.
By using the siRNA technique the authors demonstrated the role of HRAS mutation on tumor cell migration, a important factor of metastasis contributing to the suboptimal survival for patients with HRAS mutation. The authors further investigated whether HRAS knockdown or inhibition with tipifarnib affected the susceptibility of head and neck carcinoma HNC cells to cetuximab and cisplatin, two key anti-cancer agents. The findings suggest that HRAS mutations do not associate with the resistance to the current treatments. However, HRAS inhibition alone may not enhance the efficacy of standard chemotherapy in head and neck cancer.
These findings suggest that HRAS mutation testing could be a valuable tool for the development of personalized therapy among head and neck carcinoma patients. As the roles of a variety of pathways such as EGFR, PI3K, and MPK been elucidated in the ontogenetic process of head and neck carcinoma, the identification and quantification of genetic signatures using patient cohort have became an important building block toward precision cancer therapy. Guided by these genetic signature the outcomes of head and neck carcinoma patients can be improved by applying novel targeted therapies. However, given the relatively small sample size, further research and follow-up of head and neck patients with HRAS mutation with larger cohorts is necessary to validate these findings and clarify the clinical utility of HRAS-targeted therapies.
Author Response
Thank you for kind and prompt review comments.
We have updated the original manuscript according to another reviewer.
Updated points are highlighted in yellow.
We added dicussion on salivary gland carcinomas. (line 76-77, line 197-219, line 277, Table 3)
We have also updated references in accordance with update on the manuscrit (line 437-494).
We also updated superscripts / subscripts, and corrected several typos. (line 35, 58, 80, 86, 305, 319, 343, 344)
Reviewer 2 Report
Comments and Suggestions for Authors
Review of IJMS-3516366
In this manuscript, the authors examine to quantify the H-ras mutation of head and neck carcinoma, compared with clinical findings, and Knockdown of the H-ras gene for squamous cell carcinoma cell lines showed migration ability in H-ras mutant cell lines. Although this is an interesting study, this manuscript is difficult to understand because of the mixture of squamous cell carcinoma and salivary gland carcinoma. This manuscript contains some interesting and potentially useful results, however, a major revision of manuscript is needed before it can be accepted for publication.
My comments are as follows.
# This article includes salivary gland carcinoma cases, even though the main focus is on HNSCC cases. H-ras mutations in salivary gland carcinoma are found in epithelial-myoepithelial carcinoma in 30-80% of cases, and H-ras mutations have been shown to be a useful diagnostic marker for epithelial-myoepithelial carcinoma. The clinical dynamics of squamous cell carcinoma and salivary gland carcinoma are different, making this article difficult to understand. If salivary gland carcinoma is to be described, the site of origin should be divided into major and minor salivary glands. The prognosis for salivary gland carcinoma of minor salivary gland origin is worse than that of major salivary glands. Consequently, this reviewer considers that salivary gland carcinoma should not be described. If salivary gland carcinoma cases are included in the results, the authors should include a discussion of H-ras mutations and salivary gland carcinoma in the article.
# Check the subscripts and superscripts of chemical symbols and units.
Author Response
Thank you for kind and prompt review comments.
Comment 1: # This article includes salivary gland carcinoma cases, even though the main focus is on HNSCC cases. H-ras mutations in salivary gland carcinoma are found in epithelial-myoepithelial carcinoma in 30-80% of cases, and H-ras mutations have been shown to be a useful diagnostic marker for epithelial-myoepithelial carcinoma. The clinical dynamics of squamous cell carcinoma and salivary gland carcinoma are different, making this article difficult to understand. If salivary gland carcinoma is to be described, the site of origin should be divided into major and minor salivary glands. The prognosis for salivary gland carcinoma of minor salivary gland origin is worse than that of major salivary glands. Consequently, this reviewer considers that salivary gland carcinoma should not be described. If salivary gland carcinoma cases are included in the results, the authors should include a discussion of H-ras mutations and salivary gland carcinoma in the article.
Response 1: We have added site of origin to table3, with no case originated from minor salivary gland.
We have added discussion on significance on HRAS mutation on salivary gland carcinomas (line 197-219).
We have also modified introduction part focused on salivary gland carcinomas (line 76-77).
Comment 2: # Check the subscripts and superscripts of chemical symbols and units.
Response 2: We have corrected subscripts / superscripts (line 305, 319, 343, 344).
We have also updated references in accordance with update on the manuscrit (line 437-494) and corrected several typos (line 35, 58, 80, 86).
Round 2
Reviewer 2 Report
Comments and Suggestions for Authors
In this manuscript, the authors examine to quantify the H-ras mutation of head and neck carcinoma, compared with clinical findings, and Knockdown of the H-ras gene for squamous cell carcinoma cell lines showed migration ability in H-ras mutant cell lines. This manuscript interesting findings. The manuscript has been revised well. I recommend that this paper has been accepted for publication.
The authors need a little improvement.
Please correct P16 to p16.
Comments on the Quality of English LanguageNothing in particular.